# Comprehensive Evaluation System of Occupational Hazard Prevention and Control in Iron and Steel Enterprises Based on A Modified Delphi Technique

**DOI:** 10.3390/ijerph17020667

**Published:** 2020-01-20

**Authors:** Yang Song, Zhe Chen, Shengkui Zhang, Jiaojiao Wang, Chao Li, Xiaoming Li, Juxiang Yuan, Xiujun Zhang

**Affiliations:** 1School of Public Health, North China University of Science and Technology, Tangshan 063210, China; songyang@ncst.edu.cn (Y.S.); chenzhe@stu.ncst.edu.cn (Z.C.); zhangsk@stu.ncst.edu.cn (S.Z.); wangjiaojiao@stu.ncst.edu.cn (J.W.); lichao@stu.ncst.edu.cn (C.L.); lixiaoming@ncst.edu.cn (X.L.); 2Hebei Province Key Laboratory of Occupational Health and Safety for Coal Industry, North China University of Science and Technology, Tangshan 063210, China

**Keywords:** iron and steel enterprises, occupational hazards, system, Delphi method, analytic hierarchy process, fuzzy comprehensive evaluation model

## Abstract

The study designs a comprehensive evaluation system for the prevention and control of occupational hazards, calculates its weight coefficient, and provides a potential strategic and effective tool for the scientific evaluation of occupational hazards in the iron and steel enterprises. The system was established through induction and analysis of relevant literature, personal interview, theoretical analysis, Delphi expert consultation, and special group discussions. Using an improved analytical hierarchy process fuzzy comprehensive evaluation model and on the basis of the improved Delphi expert investigation, the weight of the operability comprehensive evaluation index system is constructed. A three-level index system is established on the basis of harmful factors of occupational activities, health status of employees, protection facilities of occupational hazards, occupational health management, and so on. The index system structure is 4-20-95, and the weight coefficients of the four dimensions are 0.2516, 0.2428, 0.2550, and 0.2506. The recovery rate of the questionnaire was 82.5%, 100.0%, and 100.0%. The effective rates were 75.0%, 100.0%, and 100.0%. Conversely, the expert authority coefficients of the four dimensions are 0.875, 0.769, 0.832 and 0.800. Results show that the consistency factors of the four dimensions are statistically significant. Cronbach’s α coefficient, standardized Cronbach’s α coefficient, and split-half reliability of the comprehensive evaluation index system are 0.959, 0.950, and 0.810, respectively. After factor analysis, four common factors were extracted on the basis of expert opinions, and the cumulative variance was 63.1%. The comprehensive evaluation system for the prevention and control of occupational hazards in the iron and steel enterprises proposed by the study is relatively complete and reasonable.

## 1. Introduction

The iron and steel industry is a relatively important basic industry for national economies worldwide, such as China, Japan, India, America, and Russia. Out of these countries, it is particularly not the least prominent for China [1], and relevant data have showed that the Chinese output of crude steel reached 808 million tons in 2016, which accounted for 49.6% of the worldwide production [2]. At present, China has nearly 1000 large and medium-sized iron and steel plants, in which a large number of labor employees are assiduously occupied with contributing to the booming and flourishing development of China’s iron and steel industry [3]. However, the industry can produce many occupational hazards during its production processes in combination with the large population base exposed to such occupational hazards [2,4,5,6,7,8]. This scenario is more likely to result in numerous occupational diseases and potential economic losses [9,10,11,12]. In addition, with the increasingly stringent environmental policy, especially under the growing smog pressure at the national level, the resultant prevention and control effect on occupational hazards in iron and steel foundries has become a key indicator for the evaluation of its development status [1,13]. Li suggested to implement various countermeasures to solve the problem of prevention and control of occupational diseases in Anshan from the aspects of legislation, capital investment and law enforcement [14]. Xing established an index system for the prevention and control of occupational hazards in the Bohai Bay support base of CNPC and applied it [15]. He used analytic hierarchy process, comprehensive fuzzy evaluation and Delphi based on field investigation and expert consultation to build seven indicators of occupational hazard risk factors. The assessment results showed that the hazard level was the seventh level (the lowest hazard). Zhang e used Delphi and the least squares method to construct the regression model of the occupational disease prevention performance index, and finally formed the occupational disease prevention performance system, and made a comprehensive evaluation of nine regions in Shandong Province [16]. In view of the abovementioned issues, formulating a specific system for appraising occupational hazards in iron and steel facilities in a scientific and comprehensive manner is imperative to mitigate predominant and recognized hazardous threats. The system will be of great practical significance in reducing the morbidity and mortality of occupational hazard-induced diseases [17]. However, to date, a specific study on a comprehensive evaluation system for hazardous working conditions in the Chinese context remains lacking. Therefore, based on the full consideration of the mechanisms and features of occupational hazards, the study aims to establish a comprehensive evaluation system for hazardous working environments from certain aspects, such as inherent occupational hazards, offset factors of occupational hazards and workers’ health conditions. In addition, we collected data on advanced experience from domestic and foreign assessment involving workplace occupational hazards [17,18,19,20,21]. Ultimately, the study intends to focus on formulating a comprehensive, dynamic, objective, effective as well as scientific system to mirror the status on the prevention and control of occupational hazards in the iron and steel industry based on the modified Delphi technique.

## 2. Materials and Methods

### 2.1. Selection of Consultation Experts

The selection of consultation experts plays a pivotal role when using the Delphi method to construct the index system [22], hence, we took the occurrence of unresponsiveness and withdraw into account, as well multi-disciplinary specialty across related disciplines. We invited 40 experts who are professionals engaged in the field from colleges and universities, heads, and managers of occupational health hazards from iron and steel enterprises, scholars from centers for disease control and prevention, and researchers from occupational hazard detection companies, to carry out consultation.

The study gains from the insights of 40 experts who are teachers in colleges and universities who have been engaged in relevant research for more than five years and directors and managers of occupational health in iron and steel enterprises, occupational disease prevention and control institutes, disease prevention and control centers, occupational hazard testing companies, and other relevant units. The selected number of experts is 40 to meet the minimum standard deviation under normal distribution and random sampling.

When using the Delphi method to build an index system, the selection of consulting experts plays a crucial role [22]. Therefore, we consider the occurrence of slow response and withdrawal, as well as interdisciplinary disciplines. We invited 40 university professional and technical personnel who have been engaged in relevant research for more than 5 years, the person in charge of occupational health hazards of iron and steel enterprises, managers, scholars of the Center for Disease Prevention and Control, and researchers of the company for occupational disease hazard detection for consultation. The number of experts selected is 40 to meet the minimum standard deviation under normal distribution and random sampling.

### 2.2. Selected Principles of the System

The selected principles of the comprehensive evaluation system mainly include a system with a reliable source, a system with concise operation, an all-round reflection on the various aspects of subjects, a combination of subjective and objective indicators, an integration of static and dynamic indicators, mutual independence among indicators, and a system for evaluation.

See Figure 1 for the evaluation system flow.

Figure 1 shows the whole process of the fuzzy evaluation system, and concludes that the overall situation of occupational hazard prevention and control in iron and steel enterprises needs empirical stage: the overall situation of prevention and control needs multiple levels; each level contains multiple indicators, and the weight is not easy to determine; the indicators in each level are fuzzy.

### 2.3. Selected Criterion of the System

Tutors and experts who invariably research on occupational and environmental health were consulted on the basis of the relevant literature, steel-related and other industry-related documents as well as national laws, regulations, and outline. In-depth face-to-face personal interviews were conducted on site, where data were generated through the following criteria: importance, operability, authenticity, and sensitivity, including the mean as well as the coefficient of variation (*CV*) of at least two cited indices at >7 and ≤0.25 were required. Moreover, we finally proceeded to make a primary option for the items in the system corresponding to selected principles and criteria.

### 2.4. Delphi Survey

Delphi expert consultation was utilized to collect and develop items regarding the proposed comprehensive evaluation system using the following iterative inquiry [23,24]: consultation → feedback → disposal of data → re-consultation → re-feedback → re-disposal of data. Finally, the process was repeated until the standpoints of each expert are basically in accordance with the reliability of the proposed scheme or conclusion is relatively satisfied [25].

The study was conducted for a total of three rounds of email-based Delphi survey, each of which continued for 1.5-month for total of 4.5-month (15 March 2016 to 30 July 2016). The first-round equivalent of the pilot survey of the Delphi process was used to assess the framework of system. Data such as the aim and meaning of the study, personal baseline information of experts, and a preliminary system framework, which was structured through the literature review as well as face-to-face interviews with experts. After collating and summarizing the results in round 1, the second-round items of the Delphi technique were filtered by discarding and adding items according to the proposals of the expert panel. The second-round Delphi survey primarily aims to verify and improve the system. On the basis of considering the scoring situations and opinions of the invited experts, we formulated the third-round items of the Delphi method, which was principally utilized to determine the weighting coefficients of the system during this process. Out of the three-round Delphi consultation, the invited scholars and specialists were required to rate the significance and operability of index system in round 1. A 10-point Likert-type scale was used ranging from 1 (non-significant, non-operable) to 10 (very significant, very operable) [26]. In rounds 2 and 3, the experts were expected to rate the significance, operability, sensitivity, as well as authenticity of the index system using the 10-point Likert-type scale and analytical hierarchy process (AHP) ranging from 1 (non-significant, non-operable, non-sensitive, and non-valid) to 10 (very significant, very operable, very sensitive, and very valid) [27]. Afterward, the positivity of the panelists was appraised by analyzing the recovery and response rates of the questionnaires [28]. Furthermore, the coefficients of expert authority were assessed based on the degree of familiarity as well as the quantification scores of judgment basis and influence degree for the system. The consensus coefficient of expert opinions, which is in general defined as coordination coefficient *W*, indicated the degree of consistency among the experts and consistency of scoring results during the Delphi course [29,30].

### 2.5. Determining the Weighting Coefficients for the Designed System

The weight of the comprehensive evaluation system was calculated using the AHP-fuzzy comprehensive evaluation model, which proceeds as follows [27,31,32]. Using the improved AHP method, we determine the weighting coefficients of the primary index. Then, the weight of the primary index is taken as the membership degree of the fuzzy comprehensive evaluation model. Finally, the weighting coefficients of the index is determined on the basis of fuzzy modeling.

### 2.6. Statistical Methods

The database was established using Excel 2007 software, and the reliability and validity of the index system were quantitatively evaluated using the following methods: recovery rate, response rate, Cronbach’s alpha, and factor analysis. All analyses were performed using SAS (Version 9.3; SAS Institute, Cary, NC, USA).

### 2.7. Ethical Approval

This method has been approved by the ethics committee of North China University of Science and Technology.

## 3. Results

### 3.1. Baseline Information of Consultants

At the inceptive stage, a total of 40 consultation experts were invited to participate in the study. However, only 30 questionnaires were considered valid in the third round. Table 1 shows that 30 unique panel members overlapped the course of the three-round Delphi process with 83.3% of the participants aged over 40 years old.

All experts had over 5-year length of service, out of which 66.7% experts had worked for more than 20 years. All panel members were trained with an educational background of bachelor’s degree or above, where 66.7% of the subjects obtained a graduate’s degree. All panelists had been awarded an intermediate or higher professional title, and experts who persistently concentrated their study in the field of occupational health-related work comprised 93.3% of the sample.

### 3.2. Panel Experts’ Responses to Questionnaires

A total of three rounds of the Delphi survey were conducted. A total of 40 questionnaires were distributed in round 1, out of which 33 questionnaires obtained feedback from panel members and a recovery rate of 82.5%. In the final round, only 30 questionnaires were deemed valid with a response rate of 75.5%. This rate met the requirements among respondents when three questionnaires were excluded due to incomplete or missing data. In the final two rounds, the recovery and response rates of the questionnaires reached 100%.

### 3.3. Authority Degree for Expert Panel

Table 2 shows that the expert familiarity and authority coefficients of the four main indicators are greater than 0.65 and 0.75, respectively. In general, if the authority coefficient of each index participant is greater than 0.7 [20], then the project of the index system can be regarded to be with high credibility. That is, the higher the expert authority score, the higher the project consensus proposed. Apparently, based on this argument, the projects collected in the research can be considered relatively credible with high consistency.

The expert’s familiarity with the comprehensive evaluation index system and basis for judgments are two factors that determine an expert’s authority. Familiarity with the indicator system is divided into six levels, namely, very unfamiliar, less familiar, general, more familiar, familiar, and very familiar. The basis for expert judgment of the indicator system is evaluated from four aspects, namely, theoretical analysis, practical experience, understanding, and intuition of local and abroad peers. The sums of the judgment coefficient are equal to 0.6 and 0.8, which indicate that the judgment basis has little influence on the judgment of experts and that the judgment basis has medium influence on the judgment of experts, respectively. If the sum of the discrimination coefficients is equal to 1, it means that the judgment basis has a great influence on the expert judgment.

### 3.4. Consensus in Opinions from Consultant Experts

Table 3 presents the consistency coefficients and test results of the selected items during the three rounds of the Delphi process. The table indicates that the consistency coefficients of the other indicators were statistically significant except for operability of indicators and third-party regulatory situation. In the first round of consultation, results were small with differences being statistically non-significant. Furthermore, they established a consensus on the index system to varying degrees among experts by enhancing the number of consultations, which suggests that coincident with the continuous improvement of the system and according to the remarks of the panel members, a relative coordination was observed regarding the views shared in the responses, which is necessary for the understanding of the results with a relatively high degree of credibility.

### 3.5. Retaining, Revising, or Discarding of the System During the Delphi Enquiry Process

After completing round 1, most experts decided that the items under third-party supervision and occupational health organizations and regulations should be incorporated into occupational health management as primary indicators. In total, two out of 19 secondary indicators and nine out of 95 tertiary indicators were discarded with mean scores between 6 and 7. Furthermore, three out of 19 secondary indicators and 27 out of 95 tertiary indicators were revised. Meanwhile, other participants proposed that three new secondary items and 10 novel tertiary items should be introduced to the system. In round 2, none of items were revised or deleted, whereas one new proposed item was added to the secondary indicators (data not shown). In round 3, the importance, operability, authenticity, and sensitivity of all items reached mean scores above 7 with a CV of <0.25, which indicates agreement on all items among experts and the successful formulation of the system.

### 3.6. Quantitative Determination for the Weighting Coefficients of the System

The single weighting coefficients of the comprehensive evaluation system were calculated using the formula: $P_{i}=Y_{i}/\sum_{i=1}^{j}Y_{i}$, $Y_{i}=K•W_{i}•R_{ij}•V^{\prime }$. In this study, $Y_{i}=K•W_{i}•R_{ij}•V^{\prime }$ = 1•$\left[\begin{array}{cccc} 0.3997 & 0.0359 & 0.3997 & 0.1647 \end{array}\right]•R_{ij}•\left[\begin{array}{l} 9.5\\ 7.5\\ 5.5\\ 3.5\\ 1.5 \end{array}\right]$.

Table 4 shows that we can obtain the single and hybrid weighting coefficients of the system, where the argument of *R_ij_* was denoted as a judgment matrix that corresponds to the percentage of experts who ranked a certain rate for a given number of indicators. In addition, the hybrid weighting coefficients were finally calculated using the product method based on a single weighting coefficient [21].

### 3.7. System Reliability Evaluation

For the comprehensive evaluation system, the next step of analysis is calculating the reliability coefficient (also known as structural validity), namely, Cronbach’s α and split-half reliability, which are used to measure the internal consistency of the system. The result of this coefficient is in general higher than 0.7 [17,33]. Thus, the scale is considered to have high internal consistency.

Structural validity aims to examine the relationship between the test scores and indicators. The selection of indicator data and test scores are collected simultaneously, which is the external standard for measure test effectiveness, usually the behavior we want to predict.

The expected use and advantage of the evaluation tool is that it can get a more scientific and trustworthy quantitative result by processing the fuzzy and difficult to quantify indicators through the complex digital operation of fuzzy evaluation; it can combine the qualitative and quantitative indicators organically, and the result is in line with the actual situation; it can get a vector result by processing the fuzzy comprehensive evaluation model of fuzzy mathematics, and Not a point value. In this way, it will contain rich information, which can not only accurately describe the evaluated object, but also further process and get reference information.

Table 5 provides the reliability test results of the system, which suggests that although the value of Cronbach’s α coefficient standardized by the item under personal protection in the secondary index was 0.695, Cronbach’s α coefficient and split-half reliability reached more than 0.7, which indicates that the item was considered to have an acceptable level of internal consistency. In addition, the three values of the indicators that reflected the reliability of the remainder of the items and comprehensive evaluation system were all above 0.7, which further demonstrated that these consistent items correctly reflect the field that the system aims to measure.

### 3.8. Validity of Comprehensive Evaluation System

For the questionnaire survey, validity is typically more significant than reliability [21]. In this study, validity, specifically, the effectiveness, accuracy, and correctness of the evaluation system, refers to the extent that the designed system can reflect the objective authenticity of occupational hazard prevention and control in the iron and steel enterprises.

Finally, the study aims to examine the structural validity of the 20 secondary indicators through factor analysis, which can reflect the structural validity of the evaluation system to a certain extent. Table 6 displays the elective results of factor analysis using principal component extraction for the original data. In general, factor interpretability is readily achieved with factor rotation [34]. The study was performed with equamax rotation to yield rotated factors. Observing the initial eigenvalues manifested that the common factors of the rotated and unrotated factor loadings in the respective setting were four and six, respectively, which were linked to the set of items. The eigenvalues for the first four factors were greater than 1, which accounts for 63.1% of the variance in the twenty items.

Table 7 depicts the loadings of the component matrix rotated with equamax, of which loadings greater than 0.400 are specified as strongly correlated among twenty items [31]. Thus, the study infers that the indexes of D_1_–D_11_ load highly on factor one, which indicates the factor of occupational health management. Indicators D_1_ and D_7_–D_9_ load highly on factor two, which are representative of the factor third-party supervision.

Indices A_1_–A_3_ load highly only on factor three and is indicative of the factor harmful factors in occupational activities. Indexes B_1_–B_4_ load highly only on factor four, which represents the factor workers’ health conditions. Factor five is composed of indicators D_5_ and D_10_–D_11_, which indicates the factor of occupational health organizations and regulations. Indices C_1_–C_2_ consisted mostly of factor six, which pertains to the factor protection facilities against occupational hazards. Based on the aforementioned analysis, cross-loading associations were observed among the factors. However, the study found that the loadings of items D_5_ and D_11_ in factor one was greater than that of factor five, and items D_7_–D_9_ in factor one was greater than that of factor two. Although the loadings of item D_10_ in factor five was greater than that of factor one, both reached more than 0.7 with close proximity to each other. Therefore, factors two and five can be combined as factor one, and the discriminant validity of the four primary indexes remains relatively high after merging. This result is consistent with the comments of consultants, who stated theoretical support that the reliability of the system was fairly high in the current study.

## 4. Discussion

Currently, the Delphi technique is widely applied for policy- and decision-making to reach an agreement on significant questions or opinions. The advantage of this technique is that experts without psychological pressure are not influenced by the outside environment when making judgments based on academic experience and theoretical knowledge [14,32,33]. As such, consultants can maximize their creativity to guarantee that superior viewpoints are received by putting ideas together [14]. By doing so, on the basis of the three-round Delphi survey procedure and to the best of our knowledge, the current work is the first to comprehensively build an evaluation system of occupational hazard prevention and control for the iron and steel enterprises. The study relied on literature review, synthetical analysis, field epidemiological investigation, and face-to-face interviews. During the formation of the comprehensive evaluation system, the system not only embodied the integrity, hierarchy, and rationality of occupational hazard prevention and control in the iron and steel industry, but also reflected the concept and thought of system theory, that is, the formation of the four primary indicators and development of twenty secondary indicators according to the four primary indicators to which they belonged. Finally, each secondary index was divided into a certain number of tertiary indicators of 95. The indices used to appraise the reliability of the Delphi survey mainly include numbers of consultations, numbers of panelists, representative of panelists, enthusiasm of panelists, authority as well as the consistent opinions of panelists [14,18,21]. In the present work, the respective description was presented as follows. ① For the numbers of experts: previous studies provided recommendations of the appropriate number of participants, which ranged from 15 to 50 after eliminating the underlying dropouts [14,21]. An excessive number of participants in the Delphi survey can potentially increase the burden as well as result in difficulties in terms of quality control during the consultation period. Conversely, if the participants are scarce, the formulation of the system will become unstable [34]. Thus, this study aimed to select 40 eligible scholars and specialists based on mathematical statistics theory, document literature, and eliminating possible dropouts during the survey period as respondents in the first round. A total of 33 researchers returned the questionnaires through email, where three failed to provide completed questionnaires due to busy schedules. Thus, we considered these participants irretrievable. As such, 30 subjects remained for the remainder of the survey. ② Representative of panelists: Participants with master’s degree or above selected by the study accounted for 66.7% of the sample. 93.3% of the participants had reached at least 10 years of tenure in occupational health-related professions. All of them had intermediate and above job titles. Out of the 30 subjects, 80% possessed higher job titles and all of them worked in the field of occupational health, public health, health education, epidemiology and biostatistics, health management, and health economics. In total, 93.3% of the participants are engaged in occupational health research. Furthermore, previously cited results implied that the selected experts are essential elements of good representation in terms of sound professional quality. ③ Enthusiasm of panel members: Given that the recovery and response rate of the questionnaires are equal or greater than 70%, the panelists are regarded to be with high enthusiasm and the questionnaires are of high quality [26]. The recovery and response rates of the consultation questionnaires in the first round of the Delphi process were 82.5% and 75.0%, respectively. A total of 56.7% of the respondents provided remarks for the modification of certain parts of the indicators, which indicated that the quality of the response questionnaires was considerably high, and the index system in the first round was dramatically immature. Nevertheless, the recovery and response rates of the consultation questionnaires for rounds 2 and 3 are 100.0%, where 6.7% of the consultants proposed additional comments on the index system in the second round, and none of them gave further advice in the third round. This tendency exhibits that scholars and specialists eventually acquired consensus on the items in round 2. ④ Authority of experts: A linear relationship was observed between the authority of experts and precision of the consultation results in a preceding report [34]. In the current work, the authority of the remaining three primary indicators were equal or greater than 0.8 in addition to workers’ health conditions with a value of experts’ authority calculated as 0.769. Result reveals that the experts were substantially adept in their respective research fields. ⑤ Opinion consistency of consultation experts: With the increasing number of the Delphi process, the study finds coordination among the experts’ viewpoints regarding the indicators, which were scored from four aspects, namely, importance, operability, authenticity, and sensitivity and more or less enhanced. The *W* value for indicators had a tendency to lean toward 0.5 with a significantly statistical difference in round 3. Thus, we concluded that the experts’ results in scoring the index system ultimately achieved consistency. Moreover, several large-scaled Delphi surveys implemented in the health domain concluded that the consistency coefficient *W* in the last round was generally prone to fluctuate by approximately 0.5, which indicated that the results of the present study were in accordance with domestic and foreign research [34,35].

In terms of confirming the weighting coefficients for the integrated assessment system, a modified AHP-fuzzy comprehensive evaluation methodology, which is means of empowerment along with subjective and objective combination, was adopted in the current study by referring to previous related studies [36,37,38,39]. The results were relatively congruent with theoretical and practical conditions. We find that an improved hybrid approach was ultimately utilized to adequately avert the disadvantage of the single when calculating the weighting coefficients of the system. The abovementioned result also suggested that the improved AHP-fuzzy combined method was well received as a novel model that provided investigators with not only a bulk of information regarding feasibility, rationality, and accuracy in counting the weights under study, but also a type of methodological strategy for subsequent studies.

In summary, the novel and innovatively constructed system of the study was implicated to be good feasibility, reasonability, and scientificity and can be used to comprehensively assess the preventive and control status of occupational hazards in the iron and steel industry. Moreover, it can easily identify and detect unsubstantial links responsible for the prevention and control of hazardous occupations in the iron and steel industry during the evaluation process. Finally, niche-targeting strategies for addressing this issue can be required to initiate. Notably, however, further empirical research using the comprehensive evaluation system for occupational hazards in the iron and steel industry is required to appraise the reliability and validity of the system and ultimately, improve it.

## 5. Conclusions

Based on Delphi expert consultations, the method for determining the weight of the comprehensive evaluation index system using the improved AHP-fuzzy comprehensive evaluation model is scientific and reliable. The comprehensive evaluation index system for the prevention and control of occupational hazards in the iron and steel industry is relatively inclusive, reasonable, and has high reliability and validity. In addition, the BPNN neural network model can overcome the difficulties and defects of the fuzzy comprehensive evaluation model in the empirical research stage and has a strong application prospect for comprehensive evaluation.

## Figures and Tables

**Figure 1 ijerph-17-00667-f001:**
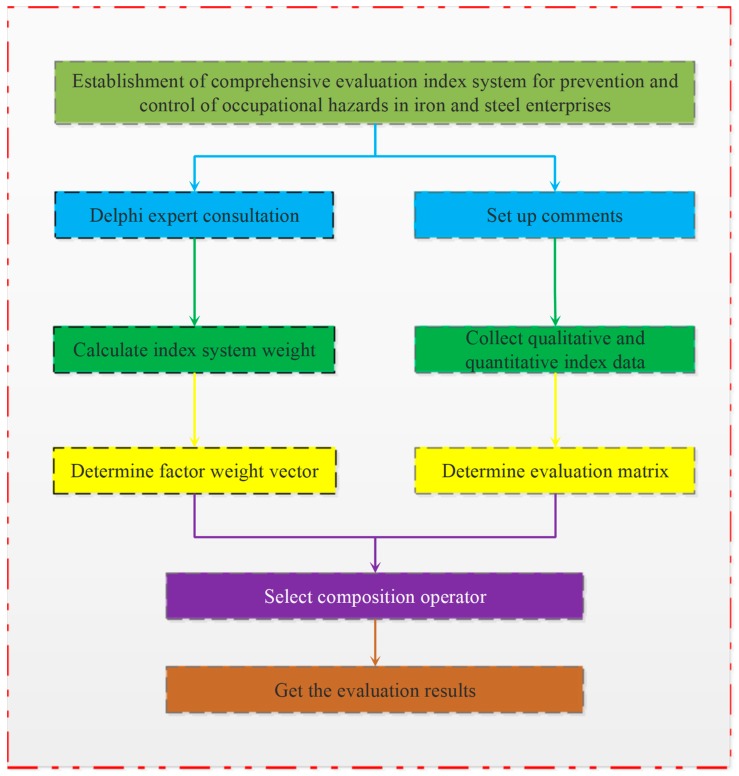
Flow chart of evaluation system.

**Table 1 ijerph-17-00667-t001:** Experts’ demographics.

Variables	Categories	N (%)
Age		
	<30	0 (0.0)
	30–39	5 (16.7)
	40–49	14 (46.7)
	50–59	10 (33.3)
	≥60	1 (3.3)
Years of working		
	5–9	2 (6.7)
	10–19	8 (26.7)
	≥20	20 (66.7)
Educational level		
	College degree or lower	0 (0.0)
	Bachelor’s degree	10 (33.3)
	Master’s degree	9 (30.0)
	Doctorate	11 (36.7)
Job title		
	Intermediate	3 (10.0)
	Vice-senior	3 (10.0)
	Senior	24 (80.0)
Professional field *		
	Health management	2 (6.7)
	Health economics	1 (3.3)
	Occupational and environmental health	28 (93.3)
	Epidemiology and biostatistics	5 (16.7)
	Public health	6 (20.0)
	Health education	3 (10.0)
	Clinical medicine	1 (3.3)
	Maternal and child health care	1 (3.3)
Main service *		
	Administrative management	9 (30.0)
	Teaching and research	22 (73.3)
	Clinical and health services	2 (6.7)

*Notes:* * This option can pertain to multiple choices, in which the proportion is defined as number of N (%).

**Table 2 ijerph-17-00667-t002:** Scores of degree of authority of expert consultation.

Items	Coefficients of Experts’ Familiarity	Judgment Coefficients	Coefficients of Authority Degree
A Harmful factors in occupational activities	0.820	0.930	0.875
B Workers’ health conditions	0.667	0.870	0.769
C Protection facilities against occupational hazards	0.747	0.917	0.832
D Occupational health management	0.727	0.873	0.800

**Table 3 ijerph-17-00667-t003:** Results of consistency and test for expert consultation.

**Rounds**	**Items**	**Importance**				**Operability**			
		***W***	$\chi_{W}^{2}$	***df***	***P***	***W***	$\chi_{W}^{2}$	***df***	***P***
First	A Harmful factors in occupational activities	0.446	280.849	21	<0.05	0.158	99.424	21	<0.05
	B Workers’ health conditions	0.235	98.500	14	<0.05	0.113	47.513	14	<0.05
	C Protection facilities against occupational hazards	0.258	116.162	15	<0.05	0.131	59.030	15	<0.05
	D Occupational health management	0.214	180.307	28	<0.05	0.182	153.070	28	<0.05
	E Third party supervision	0.105	38.553	12	<0.05	0.034	12.187	12	>0.05
	F Occupational health organizations and regulations	0.068	37.231	18	<0.05	0.060	32.396	18	<0.05
Second	A Harmful factors in occupational activities	0.429	231.740	18	<0.05	0.343	185.275	18	<0.05
	B Workers’ health conditions	0.239	100.585	14	<0.05	0.187	78.475	14	<0.05
	C Protection facilities against occupational hazards	0.409	159.996	13	<0.05	0.346	134.817	13	<0.05
	D Occupational health management	0.382	744.953	65	<0.05	0.297	579.837	65	<0.05
Third	A Harmful factors in occupational activities	0.526	284.083	18	<0.05	0.465	251.301	18	<0.05
	B Workers’ health conditions	0.308	129.103	14	<0.05	0.227	95.447	14	<0.05
	C Protection facilities against occupational hazards	0.483	188.524	13	<0.05	0.445	173.729	13	<0.05
	D Occupational health management	0.448	887.045	66	<0.05	0.395	770.598	66	<0.05
**Rounds**	**Items**	**Authenticity**				**Sensitivity**			
		***W***	$\chi_{W}^{2}$	***df***	***P***	***W***	$\chi_{W}^{2}$	***df***	***P***
Second	A Harmful factors in occupational activities	0.435	234.736	18	<0.05	0.444	239.993	18	<0.05
	B Workers’ health conditions	0.232	97.474	14	<0.05	0.201	84.450	14	<0.05
	C Protection facilities against occupational hazards	0.453	190.715	13	<0.05	0.348	135.841	13	<0.05
	D Occupational health management	0.361	703.974	65	<0.05	0.319	622.144	65	<0.05
Third	A Harmful factors in occupational activities	0.456	246.039	18	<0.05	0.414	223.443	18	<0.05
	B Workers’ health conditions	0.246	103.474	14	<0.05	0.273	114.733	14	<0.05
	C Protection facilities against occupational hazards	0.466	182.224	13	<0.05	0.389	151.178	13	<0.05
	D Occupational health management	0.374	736.290	66	<0.05	0.324	641.333	66	<0.05

*Notes*: The first round of the Delphi process failed to consult the authenticity and sensitivity of indicators.

**Table 4 ijerph-17-00667-t004:** Weighting coefficients of comprehensive evaluation systems.

Primary Indicator	Weighting Coefficient	Secondary Indicator	Weighting Coefficient	Tertiary Indicator	Weighting Coefficient	Hybrid Weighting Coefficient
A Harmful factors in occupational activities	0.2516	A_1_ Harmful factors during the production process	0.3399	A_1.1_ Qualified rate of dust monitoring points	0.1685	0.0144
				A_1.2_ Qualified rate of hazardous gas monitoring points	0.1678	0.0143
				A_1.3_ Qualified rate of high temperature monitoring points	0.1757	0.0150
				A_1.4_ Qualified rate of noise monitoring points	0.1763	0.0151
				A_1.5_ Qualified rate of thermal radiation monitoring points	0.1631	0.0140
				A_1.6_ Qualified rate of wind speed monitoring points	0.1486	0.0127
		A_2_ Harmful factors during the labor process	0.3275	A_2.1_ Labor organization and system	0.2122	0.0175
				A_2.2_ Occupational psychological stress	0.1796	0.0148
				A_2.3_ Labor intensity	0.2233	0.0184
				A_2.4_ Incorrect posture	0.1881	0.0155
				A_2.5_ Tool usage	0.1967	0.0162
		A_3_ Harmful factors in the working environment	0.3326	A_3.1_ Natural environment	0.1819	0.0152
				A_3.2_ Equipment layout	0.1947	0.0163
				A_3.3_ Daylighting and artificial lighting	0.1952	0.0163
				A_3.4_ Ventilation conditions	0.2280	0.0191
				A_3.5_ Air conditioning	0.2003	0.0168
B Workers’ health conditions	0.2428	B_1_ Workers’ basic situation	0.2485	B_1.1_ Living habits	0.3104	0.0187
				B_1.2_ Health habits	0.3212	0.0194
				B_1.3_ Average exposure length of service	0.3684	0.0222
		B_2_ Workers’ exposure characteristics	0.2596	B_2.1_ Number of exposures	0.3522	0.0222
				B_2_ B_2.2_ Level of exposure	0.3233	0.0204
				B_2.3_ Exposure time and frequency	0.3245	0.0205
		B_3_ Workers’ quality	0.2604	B_3.1_ Educational level	0.4940	0.0312
				B_3.2_ Professional skill	0.5060	0.032
		B_4_ Workers’ knowledge of, attitude toward, and practice in terms of occupational hazards	0.2315	B_4.1_ Awareness rate	0.3658	0.0206
				B_4.2_ Belief formation rate	0.2812	0.0158
				B_4.3_ Behavior formation rate	0.3530	0.0198
C Protection facilities against occupational hazards	0.2550	C_1_ Personal protection	0.4969	C_1.1_ Usage rate of personal protective equipment	0.1693	0.0214
				C_1.2_ Protective properties and performance of protective equipment	0.1700	0.0215
				C_1.3_ Routine maintenance of protective equipment	0.1685	0.0213
				C_1.4_ Maintenance replacement cycle	0.1658	0.021
				C_1.5_ Protective equipment release records	0.1571	0.0199
				C_1.6_ Appropriate types of protective equipment release	0.1693	0.0214
		C_2_ Engineering protection	0.5031	C_2.1_ Setting rate of detoxification purification facilities	0.1752	0.0225
				C_2.2_ Setting rate of noise and vibration control facility	0.1721	0.0221
				C_2.3_ Setting rate of non-ionizing radiation protection facility	0.1655	0.0212
				C_2.4_ Setting rate of heatstroke prevention, cold-proof, and moisture-proof	0.1626	0.0209
				C_2.5_ Setting rate of high temperature and thermal radiation protection facilities	0.1702	0.0218
				C_2.6_ Setting rate of other harmful factors in protection facility	0.1544	0.0198
D Occupational health management	0.2506	D_1_ Occupational health surveillance	0.0919	D_1.1_ Development of occupational health surveillance program	0.1944	0.0045
				D_1.2_ Completion rate of occupational health surveillance file	0.1984	0.0046
				D_1.3_ Examination rate of occupational health	0.2167	0.005
				D_1.4_ Physical examination of pre-service, on-the-job training, and leave	0.2126	0.0049
				D_1.5_ Post-care medical follow-up examination	0.1779	0.0041
		D_2_ Occupational hazard monitoring	0.0915	D_2.1_ Coverage rate of occupational hazard monitoring points	0.1692	0.0039
				D_2.2_ Detection rate of occupational hazard factors	0.1777	0.0041
				D_2.3_ Monitoring rate of personal dose	0.1679	0.0039
				D_2.4_ Filing and reporting of monitoring results	0.1716	0.0039
				D_2.5_ Improvement rate of supervision and inspection institutions	0.1565	0.0036
				D_2.6_ Intact rate of monitoring equipment	0.1572	0.0036
		D_3_ Materials and equipment management	0.0856	D_3.1_ Adoption of new technologies, techniques, and new materials	0.1956	0.0042
				D_3.2_ Adoption of equipment and materials for the production of occupational hazards	0.1989	0.0043
				D_3.3_ Leading suppliers of raw materials for enterprises	0.1932	0.0041
				D_3.4_ Chinese instructions of harmful equipment, hazardous chemicals, and radioactive materials	0.2052	0.0044
				D_3.5_ Public hazards of new technologies, processes, and new materials	0.2071	0.0044
		D_4_ Auxiliary health facilities	0.0822	D_4.1_ Setting rate of production health room	0.3433	0.0071
				D_4.2_ Setting rate of living room	0.3339	0.0069
				D_4.3_ Setting rate of women’s health room	0.3228	0.0066
		D_5_ Emergency rescue facilities and warning sign configuration	0.0910	D_5.1_ Establishment of emergency response agencies	0.1410	0.0032
				D_5.2_ Contingency plans	0.1419	0.0032
				D_5.3_ Relief supplies	0.1389	0.0032
				D_5.4_ Emergency drills	0.1395	0.0032
				D_5.5_ Setting rate of alarm device	0.1484	0.0034
				D_5.6_ Setting rate of warning labels	0.146	0.0033
				D_5.7_ Maintenance of emergency rescue facilities	0.1444	0.0033
		D_6_ Supervision of regulatory agencies	0.0985	D_6.1_ Rectification supervision	0.4933	0.0122
				D_6.2_ Penalties	0.5067	0.0125
		D_7_ Early prevention	0.0911	D_7.1_ Technical review and approval of construction projects	0.2562	0.0058
				D_7.2_ Declaration rate of declaration management project with occupational hazard factors	0.2436	0.0056
				D_7.3_ Occupational safety production license	0.2442	0.0056
				D_7.4_ Pre-evaluation rate of construction projects with occupational hazard factors	0.2560	0.0058
		D_8_ Protection and management during the operation process	0.0923	D_8.1_ Notification rate of occupational hazards	0.3397	0.0079
				D_8.2_ Establishment and daily management of occupational health institutions	0.3361	0.0078
				D_8.3_ Training rate of occupation health	0.3242	0.0075
		D_9_ Occupational hazard post-treatment situation	0.0881	D_9.1_ Investigating and treating rate of occupational hazard accidents	0.2530	0.0056
				D_9.2_ Qualified rate of physical examination	0.2451	0.0054
				D_9.3_ Detection rate of occupational disease	0.2487	0.0055
				D_9.4_ Coverage rate of work-related injury insurance	0.2533	0.0056
		D_10_ Occupational health organizations	0.0927	D_10.1_ Occupation health management organizations and full- or part-time management staff	0.3411	0.0079
				D_10.2_ Occupational health professional and technical monitoring institutions and specialized or concurrent vocational and technical personnel	0.3209	0.0075
				D_10.3_ Occupational health leading bodies and constituent personnel	0.3380	0.0078
		D_11_ Occupational health regulations	0.0952	D_11.1_ Management system of occupational hazard protection equipment and facilities	0.0736	0.0018
				D_11.2_ Management system of labor employment and health care files	0.0722	0.0017
				D_11.3_ Management system of the workplace	0.0704	0.0017
				D_11.4_ Management system of occupation disease diagnosis	0.0687	0.0016
				D_11.5_ Management system of occupational hazard daily monitoring	0.0719	0.0017
				D_11.6_ Evaluation system of occupational hazards	0.0706	0.0017
				D_11.7_ Notification system of occupational hazards	0.0724	0.0017
				D_11.8_ Inspection system of occupational hazards	0.0710	0.0017
				D_11.9_ Education and training system of occupational health	0.0691	0.0016
				D_11.10_ Management system of protective supplies	0.0708	0.0017
				D_11.11_ Management system of equipment maintenance	0.0717	0.0017
				D_11.12_ Declaration system of occupational disease harm	0.0716	0.0017
				D_11.13_ Evaluation system of construction projects with occupation disease harm	0.0731	0.0017
				D_11.14_ Three simultaneous management systems	0.0729	0.0017

**Table 5 ijerph-17-00667-t005:** Test results of reliability.

Items	Cronbach’s α Coefficient	Standardized Cronbach’s α Coefficient	Split-Half Reliability
A Harmful factors in occupational activities	0.824	0.823	0.782
A_1_ Harmful factors during the production process	0.707	0.705	0.742
A_2_ Harmful factors during the labor process	0.851	0.840	0.825
A_3_ Harmful factors in the working environment	0.799	0.782	0.737
B Workers’ health conditions	0.773	0.770	0.821
B_1_ Workers’ basic situation	0.793	0.788	0.717
B_2_ Workers’ contact characteristics	0.858	0.861	0.860
B_3_ Workers’ quality	0.731	0.738	0.737
B_4_ Workers’ knowledge of, attitude toward, and practice in terms of occupational hazards	0.745	0.757	0.725
C Protection facilities against occupational hazards	0.775	0.774	0.775
C_1_ Personal protection	0.714	0.695	0.740
C_2_ Engineering protection	0.871	0.919	0.881
D Occupational health management	0.765	0.783	0.746
D_1_ Occupational health surveillance	0.899	0.904	0.874
D_2_ Occupational hazard monitoring	0.895	0.903	0.827
D_3_ Materials and equipment management	0.779	0.786	0.893
D_4_ Auxiliary health facilities	0.916	0.918	0.961
D_5_ Emergency rescue facilities and warning signs configuration	0.739	0.790	0.740
D_6_ Supervision of regulatory agencies	0.887	0.887	0.887
D_7_ Early prevention	0.801	0.807	0.703
D_8_ Protection and management during operation process	0.702	0.704	0.782
D_9_ Occupational hazard post-treatment situation	0.844	0.862	0.802
D_10_ Occupational health organizations	0.707	0.769	0.769
D_11_ Occupational health regulations	0.946	0.944	0.947
Comprehensive evaluation index system	0.959	0.950	0.810

**Table 6 ijerph-17-00667-t006:** Principal component of factor analysis.

Component	Initial Eigenvalues			Extracted Sums of Squared Loadings			Rotation Sums of Squared Loadings		
	Total	% of Variance	Cumulative %	Total	% of Variance	Cumulative %	Total	% of Variance	Cumulative %
1	6.083	30.417	30.417	2.738	30.417	30.417	1.535	17.056	17.056
2	3.303	16.514	46.931	1.486	16.514	46.931	1.528	16.975	34.030
3	2.392	11.958	58.889	1.076	11.958	58.889	1.442	16.023	50.053
4	2.270	11.35	70.239	1.022	11.350	70.239	1.178	13.089	63.142
5	1.851	9.254	79.494	0.833	9.254	79.494	1.082	12.017	75.159
6	1.381	6.905	86.399	0.621	6.905	86.399	1.012	11.240	86.399
7	0.849	4.245	90.644						
8	0.506	2.530	93.174						
9	0.415	2.075	95.249						
10	0.312	1.560	96.809						
11	0.146	0.730	97.539						
12	0.118	0.590	98.129						
13	0.101	0.505	98.634						
14	0.097	0.485	99.119						
15	0.063	0.315	99.434						
16	0.049	0.245	99.679						
17	0.026	0.130	99.809						
18	0.019	0.095	99.904						
19	0.012	0.056	99.960						
20	0.008	0.040	100.000						

**Table 7 ijerph-17-00667-t007:** Component matrix rotated by equamax.

Items	Component					
	1	2	3	4	5	6
A_1_ Harmful factors during the production process	0.391	−0.096	0.497	0.305	0.155	−0.065
A_2_ Harmful factors during the labor process	−0.157	0.003	0.661	0.322	0.044	0.029
A_3_ Harmful factors in the working environment	−0.056	−0.058	0.830	0.037	0.087	0.055
B_1_ Workers’ basic situation	0.081	−0.164	0.190	0.493	0.015	−0.026
B_2_ Workers’ contact characteristics	0.035	−0.082	0.154	0.709	−0.051	−0.018
B_3_ Workers’ quality	0.113	0.168	0.180	0.817	0.095	−0.077
B_4_ Workers’ knowledge of, attitude toward, and practice in terms of occupational hazards	−0.227	0.183	−0.118	0.630	−0.119	0.280
C_1_ Personal protection	−0.038	−0.218	0.015	0.000	−0.063	0.765
C_2_ Engineering protection	0.152	0.232	−0.083	−0.046	−0.052	0.685
D_1_ Occupational health surveillance	0.661	0.426	−0.003	0.065	0.219	−0.020
D_2_ Occupational hazard monitoring	0.565	0.308	−0.043	0.248	0.074	−0.038
D_3_ Materials and equipment management	0.508	0.370	0.398	−0.015	0.320	−0.159
D_4_ Auxiliary health facilities	0.437	0.340	−0.108	−0.032	−0.096	−0.016
D_5_ Emergency rescue facilities and warning sign configuration	0.493	0.368	0.167	-0.093	0.414	0.262
D_6_ Supervision of regulatory agencies	0.532	−0.138	−0.140	0.038	0.358	0.103
D_7_ Early prevention	0.542	0.448	0.068	0.178	−0.158	0.286
D_8_ Protection and management during operation process	0.727	0.714	0.082	0.287	−0.071	0.134
D_9_ Occupational hazard post-treatment situation	0.531	0.408	−0.034	0.031	0.188	−0.120
D_10_ Occupational health organizations	0.746	0.386	0.167	−0.089	0.804	0.132
D_11_ Occupational health regulations	0.829	0.118	0.052	0.017	0.776	0.102

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
