# Peer review of "Comprehensive Evaluation System of Occupational Hazard Prevention and Control in Iron and Steel Enterprises Based on A Modified Delphi Technique"

_ijerph, 2020, doi:10.3390/ijerph17020667_

Round 1

Reviewer 1 Report

Introduction

- Author add the more literature review on the approach of occupational hazards  prevention and control in iron and steel industry

- Author add the more literature review on modified Delphi technique with the occupational hazards  prevention and control.

Materials and Methods

- Author should be showed the reason or literature support of small sample size (only 40 experts) for representative of population.

 - Author should be coded the reference of Selected principles of system and Selected principles of system parts.

- Author should be added the ethical approval.  

Result

-Author should be reconsider with the result present in Table 4 and 6, which was long table.  

Author Response

- Author add the more literature review on the approach of occupational hazards  prevention and control in iron and steel industry

In the process of setting up the index system of prevention and control of occupational disease hazards in Bohai Bay security base of PetroChina, Xing Shilei also made a detailed description of the ways of prevention and control of occupational diseases. Li Jian suggested to implement various countermeasures to solve the problem of prevention and control of occupational diseases in Anshan from the aspects of legislation, capital investment and law enforcement.

- Author add the more literature review on modified Delphi technique with the occupational hazards  prevention and control.

Xing Shilei is the index system of prevention and control of occupational hazards established by Delphi algorithm,Zhang Le used Delphi and the least square method to construct the regression model of the occupational disease prevention performance index, and finally formed the occupational disease prevention performance system, and made a comprehensive evaluation of nine regions in Shandong Province

Materials and Methods

- Author should be showed the reason or literature support of small sample size (only 40 experts) for representative of population.

The study gains from the insights of 40 experts who are teachers in colleges and universities who have been engaged in relevant research for more than five years and directors and managers of occupational health in iron and steel enterprises, occupational disease prevention and control institutes, disease prevention and control centers, occupational hazard testing companies, and other relevant units. The selected number of experts is 40 to meet the minimum standard deviation under normal distribution and random sampling.

 - Author should be coded the reference of Selected principles of system and Selected principles of system parts.

The selection principles of the comprehensive evaluation system mainly include: the system with reliable sources, the system with simple operation, the comprehensive reflection on all aspects of the subject, the combination of subjective and objective indicators, the combination of static indicators and dynamic indicators, and also the matrix calculation based on the corresponding indicator values, all of which are 1-10, which is the corresponding influence degree.

- Author should be added the ethical approval.  

This method has been approved by the ethics committee of North China University of Science and Technology.

Result

-Author should be reconsider with the result present in Table 4 and 6, which was long table.  

After checking, the data is correct.

Reviewer 2 Report

This manuscript and the study itself are very difficult to understand due to the lack of clarity in using the English language.  It is strongly suggested that use of English language/writing be edited for clarity.

Introduction:  It would be helpful to include additional information about the known safety hazards in the iron and steel industry.  It would seem that research has been done documenting such hazards, even if it has been done in a country other than China.

In the methods section, the definitions of the items that were rated by study participants (e.g., significant, operable, authenticity) in the Delphi process are not given, and their relevance to the evaluation system is not clear.  Providing definitions and describing the importance and relevance of the factors that were rated to hazard evaluation and control is strongly suggested. 

Section 3.3, Lines 155 - 160:  How were coefficients of experts' familiarity, judgment, and authority with the primary indicators obtained? What is the meaning of these terms?

What ratings were used as the basis for calculating Cronbach's alpha and split-half reliability?  To evaluate reliability of the assessment tool, it seems as though the tool would have been used to evaluate workplace hazards.  Was this done?

Can you provide the actual questions that will be used in the assessment tool? Currently, it seems as though only the central question topic has been provided rather than the questions themselves. 

Who is the intended user of the assessment tool developed in this study?  Someone external to the company being evaluated, or an internal person?

Please discuss the limitations of the study.

Several sections of the paper have not been completed, including Conclusions, Patents, Acknowledgments, and Conflicts of Interest.

Author Response

This manuscript and the study itself are very difficult to understand due to the lack of clarity in using the English language.  It is strongly suggested that use of English language/writing be edited for clarity.

The language has been modified in this paper, and the sentence with grammatical error has been corrected.

Introduction:  It would be helpful to include additional information about the known safety hazards in the iron and steel industry.  It would seem that research has been done documenting such hazards, even if it has been done in a country other than China.

About the investigation and research of occupational diseases in the iron and steel industry, there are indeed predecessors who have done or studied it. The innovation of this paper is that it uses all the intelligent algorithms related to mathematical statistics to solve the problem. The method is simple and easy to understand, with strong operability.

In the methods section, the definitions of the items that were rated by study participants (e.g., significant, operable, authenticity) in the Delphi process are not given, and their relevance to the evaluation system is not clear.  Providing definitions and describing the importance and relevance of the factors that were rated to hazard evaluation and control is strongly suggested. 

Such indicators as important, operable and authenticity are defined according to the actual situation according to the degree of influence. The defined values directly participate in the construction of the proportion matrix. These numerical matrices are used to evaluate things through AHP, and their definition levels are similar to table 1

Section 3.3, Lines 155 - 160:  How were coefficients of experts' familiarity, judgment, and authority with the primary indicators obtained? What is the meaning of these terms?

Through the questionnaire survey, after data analysis, the significance is the expert's familiarity with the occupational patients, the accuracy of disease judgment and the popularity of the disease in the medical field.

What ratings were used as the basis for calculating Cronbach's alpha and split-half reliability?  To evaluate reliability of the assessment tool, it seems as though the tool would have been used to evaluate workplace hazards.  Was this done?
Structural validity is to examine the relationship between test scores and indicators. The selection of indicator data and test scores are collected at the same time. It is the external standard to measure test effectiveness, usually the behavior we want to predict. This research tool was used in the sampling of male psychiatric patients in Brain Hospital Affiliated to Nanjing Medical University, and completed the validity evaluation. However, because the number of samples is too small, the conclusion is that the validity is not very good.

Can you provide the actual questions that will be used in the assessment tool? Currently, it seems as though only the central question topic has been provided rather than the questions themselves. 

The practical problem that this evaluation tool solves is to study whether the weight coefficient items in AHP are consistent and statistically significant.

Who is the intended user of the assessment tool developed in this study?  Someone external to the company being evaluated, or an internal person?

The prospective users of this study are doctors or companies in the medical field, and external personnel of the evaluated companies.

Please discuss the limitations of the study.

The limitation of this study is that it can solve the comprehensive evaluation of events affected by static factors. This study is not suitable for the events affected by uncertain factors such as variety or number of changes, and this study is not suitable for the events。

Several sections of the paper have not been completed, including Conclusions, Patents, Acknowledgments, and Conflicts of Interest.

From the fifth part to the seventh part of the article, the conclusion, the author's contribution and the declaration of conflict of interest are completed. Because there is no patent, there is no patent related writing.

Round 2

Reviewer 1 Report

This revsion is accepted for publication.

Author Response

Thank you very much!

Best regards!

Reviewer 2 Report

The primary concern associated with this manuscript is the lack of clarity of the steps taken by experts when generating and rating indicator items.  Please refer to Reference #26:  Blaschke, O'Callaghan, and Schofield for an example of a clear description and flow chart of the item generation and rating steps. 

Section 2.1:  This section contains redundant information and could be shortened into one paragraph by deleting the repeated information.

Lines 94 - 101:  It is not clear what data were generated by experts.  Were they numerical data? A mean greater than 7 seems to have been one of the required indices.  What is the rating scale from which means are generated?  These topics are discussed in more detail later in the manuscript. This section seems to be out of place and without complete descriptions of rating scales, information that was rated, and the data that were generated, this section is difficult to understand. 

The authors state that "data were generated through the following criteria:  importance, operability, authenticity, and sensitivity".  The first part of this phrase (data were generated through the following criteria) is unclear.  Were these criteria selected as required characteristics of the evaluation system, or was the evaluation system rated according to these criteria?  What questions were experts asked to initiate generation of the indicator items?

Lines 108 - 131:  I am assuming that experts generated the indicators listed in Table 4.  This step (i.e., experts' generation of indicator items) does not seem to be mentioned or described in the steps listed in this section.  Does "preliminary system framework" refer to the generation of indicator items?  If so, this is not clear.

Lines 120 - 125:  Please provide the definitions of significance, operability, sensitivity, and authenticity.  These terms are not clear.  Presumably, experts rated the significance, operability, sensitivity, and authenticity of the items on the index system.  How was it determined that these are important characteristics of the index system? 

Lines 127 - 131:  Who rated experts' familiarity, judgment basis, and influence degree?  Please provide the definition of these terms.

Lines 177 - 182:  Please define the terms "theoretical analysis" and "intuition".  The sentence that begins with "The sums of the judgment coefficient are equal to . . . " is not clear.  What does "which indicated that the judgment basis has little influence on the judgment" mean?  The following sentence ending with "Loud" is also unclear. 

Lines 221 - 224:  Please describe the steps used to generate the data used in the reliability analyses.

It would be helpful to include a description of the intended use and users of the evaluation tool in the manuscript.

Some of the items in the evaluation tool are not self-explanatory.  For example, what is B1. Workers' basic situation, and B2. Workers' contact characteristics?  How will such items be clearly explained to users of the evaluation tool? 

Author Response

The primary concern associated with this manuscript is the lack of clarity of the steps taken by experts when generating and rating indicator items.  Please refer to Reference #26:  Blaschke, O'Callaghan, and Schofield for an example of a clear description and flow chart of the item generation and rating steps. 

In Chapter 2.2 of the article, the process and steps of the evaluation system of occupational hazard indicators have been presented in the form of flow chart according to the idea of document 26, and the evaluation is conducted according to the steps in the flow chart in combination with the evaluation levels and indicators described in the paper

Section 2.1:  This section contains redundant information and could be shortened into one paragraph by deleting the repeated information.

This part has been modified according to the modification opinions, the redundant information has been deleted, and other contents have been combined into one paragraph.

Lines 94 - 101:  It is not clear what data were generated by experts.  Were they numerical data? A mean greater than 7 seems to have been one of the required indices.  What is the rating scale from which means are generated?  These topics are discussed in more detail later in the manuscript. This section seems to be out of place and without complete descriptions of rating scales, information that was rated, and the data that were generated, this section is difficult to understand. 

The index data of age, working years, education level, job title, professional field and main work are from these experts, some of which are numerical data and some are statistical data. If the average value is greater than 7, the chi square value can be calculated by formula, otherwise, the consistency coefficient should be obtained by looking up the table. In this part, the expert data are briefly introduced, including specific scoring standards, scoring information, etc. see Table 1 for details.

The authors state that "data were generated through the following criteria:  importance, operability, authenticity, and sensitivity".  The first part of this phrase (data were generated through the following criteria) is unclear.  Were these criteria selected as required characteristics of the evaluation system, or was the evaluation system rated according to these criteria?  What questions were experts asked to initiate generation of the indicator items?

Importance refers to the importance of all levels of factors in the indicator system, operability refers to the easy access to data through these factors, authenticity refers to the authenticity of the data obtained, sensitivity refers to the considerable impact of all levels of factors on the whole indicator system. This standard is selected according to the requirements of the evaluation system. In order to start the generation of index projects, experts asked about the situation of occupational hazards, their own situation, occupational hazard protection measures, occupational hazard management, third-party supervision, rules and regulations, etc.

Lines 108 - 131:  I am assuming that experts generated the indicators listed in Table 4.  This step (i.e., experts' generation of indicator items) does not seem to be mentioned or described in the steps listed in this section.  Does "preliminary system framework" refer to the generation of indicator items?  If so, this is not clear.

The framework in Table 4 is: the third level indicators construct the second level indicators through weighted connection, and the weighted connection between the second level indicators can form the first level. This is the final index evaluation system after several rounds of expert consultation. "Preliminary system framework" is parallel to the previous research purpose, significance and personal information of experts, and refers to the parameters and corresponding standards of the whole evaluation system.

Lines 120 - 125:  Please provide the definitions of significance, operability, sensitivity, and authenticity.  These terms are not clear.  Presumably, experts rated the significance, operability, sensitivity, and authenticity of the items on the index system.  How was it determined that these are important characteristics of the index system? 

Importance refers to whether the evaluation indexes are important for the evaluation purpose through literature review and expert consultation; operability refers to whether the indexes are convenient for collecting or obtaining data when they are actually measured; sensitivity refers to that the evaluation indexes are sensitive to the evaluation purpose and have a great impact on the evaluation purpose; authenticity refers to the nature of the data, that is, the data Accurate. To determine its important characteristics, we need to use the mean value, standard deviation and coefficient of variation, and finally we need to see its weight coefficient.

Lines 127 - 131:  Who rated experts' familiarity, judgment basis, and influence degree?  Please provide the definition of these terms.

Using the comprehensive evaluation method and statistical method, combined with the survey data of experts, the expert's familiarity, judgment basis and influence degree are determined. The expert's judgment basis for the indicator system is investigated from four aspects: theoretical analysis, practical experience, understanding and intuition of peers at home and abroad. The influence degree is defined according to the authority degree of experts.

Lines 177 - 182:  Please define the terms "theoretical analysis" and "intuition".  The sentence that begins with "The sums of the judgment coefficient are equal to . . . " is not clear.  What does "which indicated that the judgment basis has little influence on the judgment" mean?  The following sentence ending with "Loud" is also unclear. 

Theoretical analysis refers to the objective analysis of the event itself, intuition is the experts' understanding and judgment of the event, etc. The sum of judgment coefficients refers to the sum of the quantitative values of the four factors of theoretical analysis, practical experience, understanding and intuition of peers at home and abroad. There is a mistake in the last sentence of this paragraph. It is wrong to write "the sum of coefficients is equal to 1" as "1 point", which has been changed.

Lines 221 - 224:  Please describe the steps used to generate the data used in the reliability analyses.

Before the empirical study, the reliability of the comprehensive evaluation index system was evaluated by using the Cronbach A-coefficient, the Cronbach A-coefficient and the split half reliability, the structural validity and the differentiated validity of the comprehensive evaluation index system were evaluated by using factor analysis, and the comprehensive evaluation was conducted by using the recovery rate, efficiency of the questionnaire and the proportion of experts who raised questions about the index system The evaluation of the acceptability of the price index system shows that although there is a phenomenon of cross domination between factors, the analysis is based on the expert's advice and the actual situation (the third-party supervision and occupational hazard organizations and rules and regulations are essentially within the scope of occupational health management) during Delphi consultation, and the two are completely consistent, so it can be said that the research institute has made The comprehensive evaluation index system has good reliability, validity and acceptability.

It would be helpful to include a description of the intended use and users of the evaluation tool in the manuscript.

OK, it has been added in the article.

Some of the items in the evaluation tool are not self-explanatory.  For example, what is B1. Workers' basic situation, and B2. Workers' contact characteristics?  How will such items be clearly explained to users of the evaluation tool? 

The description of each project indicator is determined by the next level of indicators, for example, the basic situation of B1 workers, which is determined by the living habits of B11 workers in the next level, B12 health habits, B13 average exposure working age and their corresponding weights. For the evaluation system, the acquired data are generally the three-level index data, through which the secondary index or the primary index can be reflected.